# A Comparative Analysis of Quality of Life in Women Diagnosed with Breast and Ovarian Cancer

**DOI:** 10.3390/ijerph19116705

**Published:** 2022-05-31

**Authors:** Robert Słoniewski, Marta Dąbrowska-Bender, Urszula Religioni, Adam Fronczak, Anna Staniszewska, Aneta Duda-Zalewska, Magdalena Milewska, Magdalena Kędzierska, Rafał Adam Matkowski, Grażyna Dykowska, Anna Słoniewska, Anna Kupiecka

**Affiliations:** 1Department of Public Health, Medical University of Warsaw, 02-091 Warsaw, Poland; adamum@op.pl (A.F.); aneta.duda-zalewska@wum.edu.pl (A.D.-Z.); 2Department of Clinical Dietetics, Medical University of Warsaw, 02-091 Warsaw, Poland; magdalena.milewska@wum.edu.pl; 3School of Public Health, Centre of Postgraduate Medical Education of Warsaw, 01-813 Warsaw, Poland; urszula.religioni@gmail.com; 4Department of Experimental and Clinical Pharmacology, Medical University of Warsaw, 02-091 Warsaw, Poland; anna.staniszewska@wum.edu.pl; 5Department of Chemotherapy, Medical University of Lodz and Copernicus Memorial Hospital, CCC & T, 90-419 Lodz, Poland; kameleonmagda6@gmail.com; 6Department of Oncology, Wrocław Medical University, Lower Silesian Oncology, Pulmonology and Hematology Center, 53-413 Wroclaw, Poland; rafal.matkowski@dcopih.pl; 7Department of Health Economics and Medical Law, Medical University of Warsaw, 02-091 Warsaw, Poland; grazyna.dykowska@wum.edu.pl; 8Masovian Oncological Hospital, 05-135 Wieliszew, Poland; sloniewska.anna@gmail.com; 9OnkoCafe Foundation—“Together Better”, 04-175 Warsaw, Poland; biuro@onkocafe.pl

**Keywords:** ovarian cancer, breast cancer, quality of life, QLQ-C30 and QLQ-OV28, QLQ-BR23 questionnaire

## Abstract

Background: Previous studies showed that cancer significantly reduces the quality of life of patients. The purpose of this study was to analyze changes in the quality of life of women diagnosed with ovarian and breast cancer after surgical treatment followed by adjuvant cancer therapy. Methods: The study covered 220 women diagnosed with ovarian (*n* = 89) or breast cancer (*n* = 131) after surgical treatment followed by adjuvant cancer therapy (chemotherapy, radiotherapy, hormone therapy). The tools used to measure the patients’ quality of life were the standardized EORTC QLQ-C30 questionnaire, the QLQ-BR23 module for breast cancer and the QLQ-OV28 module for ovarian cancer. Results: The subjective assessment of the health and quality of life of the women was carried out using the EORTC QLQ-C30 questionnaire and the QLQ-OV28 and QLQ-BR23 modules. Women with breast cancer rated their health higher than women with ovarian cancer. The health assessment performed by the patients was not related to the type of cancer (*p* > 0.05). They experienced pain, dyspnea and weakness regardless of the cancer location. Moreover, women’s health status had a clinically significant impact on their family and social life, although no statistically significant differences were found between the two groups (*p* > 0.05). Whilst the patients with breast cancer rated their quality of life and health higher than the patients with ovarian cancer, the differences were not statistically significant (*p* > 0.05). Conclusions: Changes in the quality of life of women with breast and ovarian cancer concern the physical sphere, hobbies, fatigue/rest, pain, family and social spheres, and material conditions. It is necessary to support specialists at every stage of treatment of these patients, which may improve the results of the treatment and patients’ perception of health and quality of life.

## 1. Background

Ovarian cancer and breast cancer are two of the most common cancers that affect women in Europe and in the USA. Cancer has a major influence on the life of a patient. Once there is a suspicion of the disease, patients find themselves in constant fear and emotional tension. According to multiple research studies, cancer not only is a risk to health and life but also evokes strong emotional reactions such as fear, anxiety, anger and resentment [1]. The disease can contribute to the loss of a number of mental and physical abilities such as physical strength, control, social roles, items that have a special meaning to the patient, financial standing, interpersonal contacts, sexual function, physical and mental health integration, hope and mental integrity. In addition, the way the patient adapts to the disease and the type of administered cancer therapy can significantly affect the patient’s quality of life, which is an important social problem in view of the incidence of cancer and its associated deaths, course of the disease and therapy. The analysis of quality of life may provide an important source of information about health and help identify populations that face an increased risk of cancer to improve prevention.

## 2. Methods

During the months before the patients were examined, the examined patients underwent surgical treatment and then additional anti-cancer therapy (chemotherapy, radiotherapy, hormone therapy). The patients received treatment at the Oncology Centre of Lower Silesia in Wroclaw at the Department of Chemotherapy for Cancer with a Single-Day Chemotherapy Section and at the Clinic of Chemotherapy for Cancer in Lodz. Some of the patients were enrolled in the Foundation OnkoCafe—“Together better”. The criteria for including patients in the study were: informed and voluntary consent to participate in the project, diagnosis of breast and ovarian cancer by a histopathological examination, surgical treatment before the commencement of chemotherapy, radiotherapy or hormone therapy, and treatment in one of the two selected oncology centers. The exclusion criterion was the patient’s lack of consent to participate in the study.

The tools used to measure the quality of life were the standardized EORTC QLQ-C30 questionnaire (Polish edition), the QLQ-BR23 module for breast cancer and the QLQ-OV28 module for ovarian cancer. The EORTC QLQ C30 questionnaire (rev. 3.0) was developed by the European Organization for Research and Treatment of Cancer (EORTC). The scale was adapted by EORTC to more than 80 languages, including Polish. All the above-mentioned scales are coded to give a score from 0 to 100, where the highest value on the scale means the highest intensity of the analyzed feature. In addition, the study was approved by the Bioethics Committee of the Medical University of Warsaw (no AKBE/32/16). All the women involved in the study gave an informed and voluntary consent to participate in it. Research evidence was gathered in the period June 2020–May 2021. The comparison of selected aspects of quality of life in the two independent groups (patients with breast cancer and patients with ovarian cancer) was made through the Mann–Whitney U test. The adopted statistical significance level was *p* < 0.05.

## 3. Results

### 3.1. Characteristics of the Population

The study covered a group of 220 patients (89 women diagnosed with ovarian cancer and 131 women diagnosed with breast cancer). The mean age of the patients was 55.52 years: 56.34 years for women with ovarian cancer and 54.96 years for women with breast cancer. Patients aged 52 prevailed in the population.

The population was divided into nine age groups with a 5-year span in age. The largest group was that composed of patients aged >65 years (25.45%), the smallest group was that of patients under 30–34 years of age (1.81%). The group of patients aged over 65 was the largest for both ovarian cancer (30.34%) and breast cancer (22.14%). The largest group of women (36.81%) lived in cities with a population of over 500,000, 28.19% of the women lived in cities with a population of up to 50,000, 18.18% of the women lived in the countryside, 9.09% in cities of 50,000 to 100,000 inhabitants, and 7.73% in cities of 100,000 to 500,000 inhabitants. More than a half of the women were married (56.36%), while 19.09% were widows, and 12.27% were single or divorced. When it comes to their professional status, 39.54% of the women were pensioners, 37.73% were employed, 14.55% were unemployed, and 8.18% were students. More than a half of the women were receiving cancer therapy (75.45%) at the time of the study. Nearly 9% of the patients were awaiting therapy, others had already completed cancer therapy. The patients were also asked about the type of therapy they had received, were receiving or would receive. The largest group was administered chemotherapy (74.54%), followed by groups that had surgery (50.00%), radiotherapy (33.18%) and hormone therapy (23.64%). In all, 27.27% of the patients pointed to a significant deterioration in their health as a result of cancer.

During the study, 16.36% of the women were pre-menopausal, 17.73% were in the course of menopause, and 65.91% were post-menopausal.

### 3.2. General Subjective Assessment of Health and Quality of Life Made by the Patients

The patients were asked to make a subjective assessment of their health in the last week (on a scale of 1 to 7, where 1 means very poor health, and 7 means excellent health). The largest group of patients rated their health at 4 (36.36%), 22.27% rated it at 5, and 13.18% rated it at 6. In addition, 5.00% of the women declared their health to be excellent, and 4.54% very poor.

The mean health rate in the population was 4.23, with a standard deviation of 1.41. The patients with breast cancer rated their health higher (4.39) in comparison to the patients with ovarian cancer (4.00). Nonetheless, the subjective health assessment made by the patients was not related to the type of cancer (*p* > 0.05).

The patients were also asked to make a subjective assessment of their quality of life in the last week (on a scale of 1 to 7, where 1 means very low quality of life, and 7 means excellent quality of life). The largest group of patients rated their quality of life at 4 (33.64%), 20.91% rated it at 5, and 15.91% rated it at 3. In addition, 5.91% of the women evaluated their quality of life to be either excellent or very low (Table 1).

The mean quality of life rate in the population was 4.15, with a standard deviation of 1.46. Whilst the patients with breast cancer rated their quality of life higher than the patients with ovarian cancer (4.31 and 3.91, respectively), the differences were not statistically significant (*p* > 0.05).

### 3.3. Physical Functioning

In terms of physical functioning, the women were asked about problems with tiring activities, fatigue during long and short walks, need for help during daily activities, limitations in performing daily activities, limitations in pursuing their hobbies, need to rest during the day and the sense of fatigue. These aspects were not determined by the type of disease—the patients with ovarian cancer and the patients with breast cancer gave similar answers. Overall, 86% of the patients had problems with tiring activities; one in four women said those problems were very large, and nearly 26% said they were significant. More than 45% of the women said they suffered from fatigue during short walks, and over 77% of the patients declared that they felt tired during long walks. The need for help during daily activities was reported by 15.91% of the patients, and 1.82% evaluated it to be very large. Nearly 70% of the women said that they encountered limitations in performing daily work or activities, and a slightly smaller number of patients (67.27%) experienced limitations in pursuing their hobbies. For nearly 90% of the women, it was essential to rest during the day, with 11.82% stating that the need for rest was very large.

Only 13.64% of the patients did not experience fatigue during the day, with 56.36% suffering from slight fatigue, 21.36% from significant fatigue, and 8.64% from severe fatigue (Table 2).

### 3.4. Pain

In terms of pain, the patients were asked about pain sensation as well as dyspnea, weakness and pain that made it difficult to perform daily activities. No statistically significant relationships were found in these aspects between the intensity of pain and the location of the tumor. Nearly 70% of the women reported that they experienced pain in the last week, with 7.27% stating that they experienced this problem very often. More than 42% of the women suffered from dyspnea, and nearly 80% from weakness. Pain that made it difficult to perform daily activities was reported by 56.36% of the patients. Table 3 and Table 4 present the characteristics of the above variables.

### 3.5. Family and Social Functioning

The patients were also asked to evaluate how their health affected their family and social life and financial standing. By correlating each of the above factors with the location of the tumor, it was revealed that the location of the tumor did not significantly affect family and social life in the population of women. Only 33.64% of the patients stated that their health had no impact at all on their family life, and 28.18% saw no connection between their health and problems in social life. However, more than 26.36% of the women indicated that their health made their family life significantly or much more difficult, and 33.64% of the patients declared the existence of significant or very large problems in social life. More than 66% of the patients admitted that their health was a cause of financial problems, with nearly 30% pointing to a significant or very high correlation (Table 5).

## 4. Discussion

This study that aimed to evaluate the quality of life of women with breast cancer and ovarian cancer by means of the EORTC QLQ-C30 questionnaire (rev. 3.0) revealed similar results for the examined women in terms of subjective assessment of health and quality of life and in aspects of physical functioning, pain and family and social life.

The patients with breast cancer rated their health slightly higher (mean value of 4.39) than the patients with ovarian cancer (mean value of 4.00). Similarly, the patients with breast cancer rated their quality of life higher (mean value of 4.31) than the patients with ovarian cancer (mean value of 3.91), although none of these differences were statistically significant (*p* > 0.05). The groups of women pointed to the existence of serious problems in performing tiring activities (86.92%) and during long walks (77.27%), limitations in performing daily work or activities (69.09%) and limitations in pursing their hobbies (67.27%). The women declared that they suffered from fatigue (86.36%) and needed to rest during the day (89.55%).

Other authors who obtained similar results in terms of the absence of differences between patients with particular types of cancer were, e.g., Greimel et al. [1] who analyzed the subjective health assessment made by women with various types of cancer. In contrast, the quality of life of male patients appeared to be slightly higher, for instance, in the study of Dąbrowska-Bender et al. [2], in which the quality of life among men with prostate cancer was rated at 4.79.

We found that 68.64% of the women experienced pain, without any relation to the location of their tumor (*p* > 0.05). Dyspnea and weakness occurred at a similar rate.

It has been estimated that more than half of cancer patients experience pain; at the terminal stage of the disease, this percentage can reach 80–90% [3,4]. Pain can cause discomfort in performing daily activities and may significantly reduce the health-dependent quality of life, which is why a lot of attention is given to the physical as well as mental aspects of it. For instance, Vaz et al. [5] demonstrated that pain experienced by women with cervical cancer or endometrial cancer significantly reduced their quality of life and self-assessed health.

According to research, patients with the same primary tumor may experience pain of different intensities and respond differently to administered painkillers [6]. Psychological factors, e.g., the ability to cope with emotions or the attitude towards pain and disease, clearly affect the intensity of the experienced pain [7,8]. Adequate pain- (physical and mental) coping strategies can reduce the sensation of pain and therefore improve the subjective assessment of quality of life [9]. Kazalska [10] stresses that the intensification of the experienced pain is largely dependent on psychological factors that play an essential role in selecting the method to cope with a chronic illness. The selection of the method of coping with a disease and the way of adapting to a disease have a significant impact on the quality of life of patients. For instance, Johansson et al. [11] and Thome and Halberg [12] found that patients who feel a high degree of helplessness and suffer from anxiety rate their quality of life much lower.

In this study, women with both breast cancer and ovarian cancer declared that their health significantly impacted on their family and social life (no statistically significant differences were found between the groups, *p* > 0.05). We found that 66.36% of the women reported problems in family life, and 71.82% problems in social life. For 66.36% of the patients, their health condition affected their financial standing.

In this context, Ogińska-Bulik [13] noted that the acceptance of a disease and therefore the reduced intensity of negative emotions associated with it and the recognition of the limitations that it entails are determined, e.g., by the intensity of the experienced symptoms. Dijkstra et al. [14] and Telford et al. [15] reached similar conclusions.

Interestingly, the patients with breast cancer and reproductive system cancers have a similar attitude towards the disease. Both Czerw et al. [16] and Kozak [17] pointed out that most of the women showed a fighting spirit. and destructive strategies such as helplessness and hopelessness were rare. Patients with other types of cancer, e.g., stomach cancer, pancreatic cancer, colorectal cancer or prostate cancer, had a different way of coping with the disease [17]. The prevalence of constructive coping strategies among women with breast cancer was also noted by Szczepańska-Gieracha et al. [18] The above-mentioned results and the results of the authors’ study suggest that women with breast cancer or reproductive system cancers have a similar response to symptoms, therapy and limitations imposed on them by their disease.

There is a large number of research studies that indicate that the attitude to cancer, including the subjective assessment of quality of life, depends on the time that passed since the diagnosis, the stage of therapy and the stage of cancer [17,19,20].

The improvement of the quality of life of cancer patients, including psychological and social support to patients and their families, is becoming an important challenge for doctors and psycho-oncologists. A large number of research studies clearly showed that the quality of life of cancer patients very much affects survivorship among them [21,22,23].

## 5. Conclusions

This study showed that changes in the quality of life of women with breast and ovarian cancer mainly concerned the physical sphere, hobbies, fatigue/rest, pain, family and social spheres, and material status. There is an urgent need for technical support, psycho-oncologists’ support, medical support (including helping patients with other types of cancer) to improve the health and quality of life of these patients.

## Figures and Tables

**Table 1 ijerph-19-06705-t001:** Quality of life assessment made by the patients (N = 220).

Quality of Life Assessment	Total Number of Patients (N = 220) N (%)
1	13 (5.91)
2	14 (6.36)
3	35 (15.91)
4	74 (33.64)
5	46 (20.91)
6	35 (11.36)
7	13 (5.91)

**Table 2 ijerph-19-06705-t002:** Assessment of physical functioning made by the patients.

Physical Activity	Number of Questions ^1^	Number of Patients (N = 220)N (%)			
		Not at All	A Little	Quite a Bit	Very Much
Do you have any trouble doing strenuous activities, like carrying a heavy shopping bag or a suitcase?	1	29 (13.18)	79 (35.91)	57 (25.91)	55 (25.00)
Do you have any trouble taking a long walk?	2	50 (22.73)	93 (42.27)	43 (19.55)	34 (15.45)
Do you have any trouble taking a short walk outside of the house?	3	142 (64.54)	50 (22.73)	20 (9.09)	8 (3.64)
Do you need help with eating, dressing, washing yourself or using the toilet?	5	185 (84.09)	25 (11.36)	6 (2.73)	4 (1.82)
Were you limited in doing either your work or other daily activities?	6	68 (30.91)	100 (45.45)	35 (15.91)	17 (7.73)
Were you limited in pursuing your hobbies or other leisure time activities?	7	72 (32.73)	84 (38.18)	49 (22.27)	15 (6.82)
Did you need to rest?	10	23 (10.45)	111 (50.45)	60 (27.27)	26 (11.82)
Were you tired?	18	30 (13.64)	124 (56.36)	47 (21.36)	19 (8.64)

^1^ Item number according to the QLQ-C30 questionnaire.

**Table 3 ijerph-19-06705-t003:** The impact of the location of the tumor (ovaries, breasts) on selected aspects of physical functioning.

Physical Activity	Number of Questions ^1^	*p*-Value
Do you have any trouble doing strenuous activities, like carrying a heavy shopping bag or a suitcase?	1	0.696
Do you have any trouble taking a long walk?	2	0.876
Do you have any trouble taking a short walk outside of the house?	3	0.346
Do you need help with eating, dressing, washing yourself or using the toilet?	5	0.399
Were you limited in doing either your work or other daily activities?	6	0.255
Were you limited in pursuing your hobbies or other leisure time activities?	7	0.376
Did you need to rest?	10	0.749
Were you tired?	18	0.312

^1^ Item number according to the QLQ-C30 questionnaire.

**Table 4 ijerph-19-06705-t004:** Assessment of pain made by the patients.

Pain	Number of Questions ^1^	Number of Patients (N = 220)N (%)			
		Not at All	A Little	Quite a Bit	Very Much
Have you had pain?	9	69 (31.36)	95 (43.18)	40 (18.18)	16 (7.27)
Were you short of breath?	8	127 (57.73)	62 (28.18)	19 (8.64)	12 (5.45)
Have you felt weak?	12	43 (19.55)	104 (47.27)	48 (21.82)	25 (11.36)
Did pain interfere with your daily activities?	19	96 (43.64)	83 (37.73)	25 (11.36)	16 (7.27)

^1^ Item number according to the QLQ-C30 questionnaire.

**Table 5 ijerph-19-06705-t005:** Assessment of family and social functioning made by the patients.

Family and Social Functioning	Number of Questions ^1^	Number of Patients (N = 220)N (%)			
		Not at All	A Little	Quite a Bit	Very Much
Has your physical condition or medical treatment interfered with your family life?	26	74 (33.64)	88 (40.00)	40 (18.18)	18 (8.18)
Has your physical condition or medical treatment interfered with your social activities?	27	62 (28.18)	84 (38.18)	44 (20.00)	30 (13.64)
Has your physical condition or medical treatment caused you financial difficulties?	28	74 (33.64)	82 (37.27)	40 (18.18)	24 (10.91)

^1^ Item number according to the QLQ-C30 questionnaire.

## Data Availability

All data are available from the corresponding author.

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
