# Peer review of "A Comparative Analysis of Quality of Life in Women Diagnosed with Breast and Ovarian Cancer"

_ijerph, 2022, doi:10.3390/ijerph19116705_

Round 1

Reviewer 1 Report

the attached article provides a lot of recommendations

to enhance the quality and relevance of the content

Author Response

Thank you very much for the suggestions sent, which certainly greatly influenced the quality of our manuscript. Due to the difficulty of responding to each of the suggestions directly in the text, we encourage you to read the new version of the work, taking into account the instructions of all reviewers. Thank you very much again.

Reviewer 2 Report

In the work entitled A comparative analysis of quality of life of women diagnosed with breast and ovarian cancer, the Authors raise an EXTREMELY IMPORTANT problem of the quality of life of cancer patients.
The studies examined the standard of living of over 200 patients who were undergoing gynecological cancer treatment. The researchers asked about the general well-being, physical activity, independence and the level of pain experienced by the patients. They were also interested in their emotions and relations with their family and society.
The study was carried out meticulously and the results are clearly presented in the table and properly prepared.
The research clearly shows that the standard of living of cancer patients definitely needs to be improved. Research, such as SÅ‚owniewski's, proves that patients need not only effective anti-cancer therapy, but also supportive treatment, eliminating pain and improving mental health. Psychological support of patients and their families is necessary for a dignified undergoing through the disease and can definitely improve the response to basic treatment.
I congratulate the Authors on their sensitivity and commitment, I hope that your research will be continued. I believe that it would be worth dividing the patients according to their age, menopausal, professional and family status. I am curious if considering the patients' pre-disease lifestyle will have an impact on the results.

Author Response

Dear Reviewer
We would like to thank you for the positive review of our article and for the perception of the issue / intervention we have raised as very important in the scope of comprehensive oncological patient care. In our future research, we will consider the proposed division of patients into age, menopausal status, occupation, and having a family / children. We will also take into account the lifestyle of patients before diagnosis of cancer and its impact on quality of life and the results of cancer treatment.

Reviewer 3 Report

The study that aimed to evaluate the quality of life of women with breast cancer and ovarian cancer by means of the EORTC QLQ-C30 questionnaire (rev. 3.0) revealed that the women had similar results in terms of the subjective assessment of health and quality of life and in aspects of physical functioning, pain and family and social life. I do have some comments as listed below in the order noted.

Comment 1: 

The quality of the data set is very important, especially for 220 patients diagnosed with ovarian cancer or breast cancer who underwent surgical treatment followed by adjuvant cancer therapy. For this reason, please clarify the inclusion criteria and exclusion criteria of sample collection in the Methods section and please also provide a flowchart immediately at the section of data collection.

Comment 2:

Please provide the norms of each dimension of the QLQ-C30 in Poland and compare the differences of each dimension of the QLQ-C30 between the norms and the study population in the present study in Tables 2, 4 & 5.

Comment 3:

Please also provide the score of each dimension of the QLQ-C30 between ovaries and breasts in Table 3.

Author Response

Dear Reviewer,
Thank you for sending your comments and suggestions for our study.

Reply to comment 1.
The criteria for including patients in the study are: informed and voluntary consent to participate in the project, diagnosis of breast and ovarian cancer in a histopathological examination, surgical treatment before the commencement of chemotherapy, radiotherapy or hormone therapy, and treatment in one of the two selected oncology centers. On the other hand, the criteria for excluding patients from the study are opposite to the inclusion criteria.

Reply to comment 2. 

Thank you very much for this comment. Due to the very similar results of both groups and the lack of dependence, it seems to us that it can be much more valuable to show the results of each area broken down into the frequency of giving a specific answer. We hope that such a presentation of the results will be much more interesting for the readers.   Reply to comment 3.  As far as we know, no standards have been established in Poland for these specific groups of patients. We presented the comparison of the obtained values with the results of other authors in the discussion. We hope this section provides extensive references to other studies.  

Round 2

Reviewer 1 Report

the criteria for excluding patients from the study are opposite to the inclusion criteria.

THIS IS NOT CLEAR AND THE AUTHORS SHOULD BE MORE PRECISE WITH THIS SENTENCE

PLEASE NOTE THE BOTH CONCLUSIOINS ARE SLIGHTLY DIFFERENT

THE ABSTRACT INCLUDES CULTURE

PLEASE REVISE

Reviewer 3 Report

The authors have replied to the reviewer's comments completely. I have no comments. I suggest that this paper can be accepted for publication.
